# Evaluation of the Local Immune Response to Hydatid Cysts in Sheep Liver

**DOI:** 10.3390/vetsci10050315

**Published:** 2023-04-27

**Authors:** Davide De Biase, Francesco Prisco, Paola Pepe, Antonio Bosco, Giuseppe Piegari, Ilaria d’Aquino, Valeria Russo, Serenella Papparella, Maria Paola Maurelli, Laura Rinaldi, Orlando Paciello

**Affiliations:** 1Department of Pharmacy, University of Salerno, 84084 Fisciano, Italy; 2Department of Veterinary Medicine and Animal Production, University of Napoli “Federico II”, CREMOPAR, Via Delpino, 1, 80137 Napoli, Italy

**Keywords:** *Echinococcus granulosus*, cystic echinococcosis, sheep, hydatid cyst, immunopathology

## Abstract

**Simple Summary:**

Cystic echinococcosis (CE) is a zoonotic parasitic disease caused by the tapeworm *Echinococcus granulosus*. Dogs and other carnivores represent the definitive hosts in the life cycle of this parasite, whereas herbivores/omnivores, including humans, are the intermediate hosts, where the larval stage (metacestode) develops in the organs (mainly liver and lungs). Among the intermediate hosts, sheep have recently gained more attention as reservoirs of infection by *E. granulosus*, but there is still poor information about the local inflammatory response associated with liver cystic echinococcosis. With our study, we aimed to contribute to “finding the gaps” and further define the immunological mechanisms involved during the different stages of the development of ovine hydatidosis.

**Abstract:**

In order to characterize the inflammatory phenotype of livers of sheep naturally infected by cystic echinococcosis, 100 sheep livers have been macroscopically assessed for the presence of hydatid cysts and sampled for histopathological and molecular analysis. According to gross and microscopic examination, livers were subsequently classified into three groups: normal liver (Group A), liver with the presence of fertile hydatid cysts (Group B), and liver with the presence of sterile hydatid cysts (Group C). Immunohistochemical analyses were accomplished using primary antibodies anti-Iba1, anti-CD3, anti-CD20, anti-TGF-β, and anti-MMP9. Finally, real-time PCR was performed in order to estimate the concentration levels of tumor necrosis factor-α (TNF-α), interferon-γ (INF-γ), interleukin (IL)-12, IL-10, and TGF-β. Immunohistochemical analysis showed a diffuse immunolabelling of mononuclear cells for Iba-1 and TGF-β and a higher amount of CD20+ B cells compared to CD3+ T cells in both Groups B and C. The expression levels of Th-1-like immune cytokines TNF-α, INF-γ, and IL-12 did not show significant statistical differences. However, we found a significant increase in expression levels of Th-2 immune cytokines TGF-β and IL-10 in Groups B and C compared to Group A. Taken together, our findings suggest that macrophages have a predominant role in the local immune response to cystic echinococcosis. Moreover, we can speculate that Th2 immunity may be dominant, corroborating the idea that B cells are decisively essential in the control of the immune response during parasitic infection and that the immunomodulatory role of IL-10 and TGF-β may ensure the persistence of the parasite within the host.

## 1. Introduction

Cystic echinococcosis (CE) is a zoonotic parasitic disease caused by larval stages (metacestodes) of the tapeworm *Echinococcus granulosus* [1,2]. The adult cestode resides in the small intestine of carnivore-definitive hosts, including dogs and other canids, and produces eggs containing the infective *oncosphere* [3]. Oncospheres are ingested by the intermediate host, and, subsequently, the metacestode develops in the viscera (mostly the liver and lungs) as a fluid-filled cyst [3]. Humans may be inadvertently infected as intermediate hosts mostly by direct contact with infected dog feces or by contaminated food or water [4,5]. *Echinococcus granulosus* cyst has a very complex organization and its structure consists of a parasitic (hydatid) and a host-derived (adventitia) component [6]. The parts originating from the parasite are composed of two layers: an inner nucleated layer that produces protoscoleces (PSC), brood capsules, daughter vesicles, and hydatid cyst fluid (HCF), known as *germinal layer,* and an outer acellular layer, known as *laminated layer* [5]. Germinal and laminated layers are enclosed by a *fibrous capsule* produced by the host [5]. The hydatid cyst may be considered fertile or sterile based, respectively, on the presence or absence of *protoscoleces* within the cyst or in the cyst wall. The immune response to *E. granulosus* infection has been classically divided into two different moments, pre-encystment and post-encystment phases, which differ for the formation of the laminated layer around the developing infective oncospheres [7]. Several authors reported that, in intermediate hosts, the early-establishment-phase cysts stimulate a Th1-type immune response that is possibly responsible for the elimination of most of the infective parasites also inducing high levels of protection against a subsequent challenge [8]. However, the immune interaction between the host and the parasite is multifaceted, encompassing effective parasite-killing immune mechanisms modulated by the parasite, which in turn are implemented by the host [9,10]. The intermediate host and the well-established hydatid cysts may co-habit for a long time, mostly with the absence of clinical symptoms and poor to moderate inflammation [9]. In human and animal models of CE, a widely accepted scenario suggests that the typical Th2-type response plays a fundamental role in the established *Echinococcus* cystic stage involving the cytokines IL-4, IL-5, IL-10, and IL-13 and a mixed population of inflammatory cells, such as eosinophils, mast cells, alternatively activated macrophages, lymphocytes, and plasma cells [10,11]. Nevertheless, the exact role of Th2 responses in parasitic infections is currently under investigation; *E. granulosus* is likely capable to control the interaction between cells of the immune system through the release of antigens that induce a Th2 response and the downregulation of regulatory T and B cells [11]. Moreover, it is speculated that Th2-type immune response may be not only responsible in the determination of the parasitic infection but also may be meaningfully associated with chronic infection [11]. Among the intermediate hosts, sheep have recently gained more attention as reservoirs of infection by *E. granulosus* [12], but studies investigating the local response to established hydatid cysts in ovine hosts are still lacking [13]. With these premises, our study aimed to further characterize the cellular inflammatory populations and cytokines expression in the ovine liver with chronic CE. For this purpose, a morphological and immunohistochemical analysis was accomplished to quantify the inflammatory cell infiltrates; specifically, a panel of mono/polyclonal antibodies against CD3, CD20, Iba1, TGF-β, and MMP9 was used to identify the different cells, cytokines, and enzymes responsible for the inflammation, tissue repair, and fibrosis. Moreover, the mRNA expression levels of tumor necrosis factor-α (TNF-α), interferon-γ (INF-γ), interleukin (IL)-12, IL-10, and TGF-β were estimated by real-time PCR to further assess the inflammatory “*scenario*” at the periphery of fertile and sterile hydatid cysts.

## 2. Materials and Methods

### 2.1. Animal Selection

For this study, we carried out morphological, immunohistochemical, and molecular analyses on 100 female sheep of different breeds, belonging to farms located in an area highly endemic for CE of southern Italy [14,15] and scheduled for slaughter. Inclusion criteria for animal selection comprised anamnestic data indicating the presence of positive animals to CE, the accessibility to pastures shared with wild animals, and the presence of shepherd dogs within the flock [16]. Animals went through a complete physical examination by which any apparent clinical illness was excluded. In addition, the absence of prion diseases was confirmed in all animals by the rapid test as recommended by European law. 

### 2.2. Ethic Statement

The study did not require consent or ethical approval according to European Directive 2010/63/EU because all sampling procedures from animals were performed during post mortem inspection. Nevertheless, the animals were slaughtered in strict accordance with European slaughter regulations (CE no: 1099/2009 of 24 September 2009) that guarantee the protection of animals at the time of the killing. Experimental protocols received institutional approval from the Ethical Animal Care and Use Committee of the University of Naples Federico II (Protocol No. PG/2021/0058962). The owner of the abattoir and the veterinary inspector responsible for the sanitary surveillance granted permission to collect the samples. 

### 2.3. Liver Macroscopic Examination

Based on the macroscopic examination, livers with no cysts, other hepatic-parasite-related lesions, or significant pathologic alterations were considered normal controls (Group A). Livers containing hydatid cysts were collected and subsequently distributed into two groups according to the fertility of the cysts (Group B, liver with fertile hydatid cysts, and Group C, liver with sterile hydatid cysts). A previously described protocol by Mathewos et al. [17] was used to determine the fertility of each cyst. Briefly, a sterile hypodermic needle was used to reduce the pressure of the cystic fluid. Then, the cyst was incised with a sterile blade and the contents were emptied into a Petri dish for examination. The presence of cystic fluid or protoscolices appearing as white spots on the germinal epithelium was considered an indication of fertility. Sterile cysts were characterized by the absence of protoscolices and a cloudy fluid-filled cavity; moreover, calcified sterile cysts generated a gritty substance when incised. Representative liver samples, associated with hydatid cysts, from each animal were collected immediately after the slaughter and were divided into two aliquots. The first aliquot was well-preserved in 10% neutral buffered formalin (code no. 05-01007Q, Bio-Optica, Milan, Italy) for histopathological examinations, and the second one was stored at −20 °C for molecular analysis. 

### 2.4. Histopathological Examination

Formalin-fixed and paraffin-embedded liver samples were cut into 4 μm thick sections and stained with hematoxylin and eosin for morphology. Masson’s trichrome staining was performed to establish the grade of fibrosis. The degree of inflammation was semi-quantitatively scored according to the ratio between the severity of inflammatory infiltrate and the area examined, as follows: 0 (absent), 1 (mild), 2 (moderate), and 3 (severe inflammation) [18]. For the severity of the inflammation, the ratio was estimated by observing at least 10 fields at 40× magnification per animal. The degree of fibrosis was graded according to the ratio between fibrosis and the area examined, as follows: 0 (absent), 1 (mild; <10%), 2 (moderate; 10–30%), and 3 (severe; >30%) [18]. For the degree of fibrosis, the ratio was estimated by observing at least 10 fields at 200× magnification.

### 2.5. Immunohistochemistry

Immunohistochemical staining for evaluating inflammatory infiltrate was performed using a well-established protocol described elsewhere [19]. Briefly, 4 μm thick sections of liver were placed on positively charged glass slides (Bio-Optica, Milan, Italy). For antigen retrieval, a pretreatment was created using a heat-induced epitope retrieval (HIER) citrate buffer pH 6.0 (Bio-Optica, Milan, Italy) for 20 min at 98 °C. Following, endogenous peroxidase (EP) activity was doused by applying 3% hydrogen peroxide (H_2_O_2_) block for 15 min at room temperature, and then the sections were incubated for 30 min with a protein block (Biocare Medical LLC). The primary antibodies were diluted in phosphate-buffered saline (0.01 M PBS, pH 7.2) and incubated overnight at 4 °C. Primary antibodies used for this study are summarized in Table 1. Horseradish peroxidase (HRP) polymer was added for 30 min at room temperature and antigen–antibody reaction was visualized using the 3,3′-diaminobenzidine (DAB) chromogen diluted in DAB substrate buffer. Finally, the slides were counterstained with hematoxylin. Between all incubation steps, slides were washed two times (5 min each) in PBS. To test the specificity of staining and according to the most recent and relevant guidelines [20], in the corresponding negative control sections, the primary antibody was either omitted or replaced with an irrelevant and unspecific IgG. The inflammatory cell phenotypes were determined according to the staining pattern of antibodies against cell surface proteins. The results of immunohistochemical staining were evaluated semi-quantitatively by counting the number of immunolabeled cells in 10 fields randomly selected with a light microscope at x400 magnification. Immunohistochemical scoring was independently performed by two pathologists (OP and DDB) with a concordance rate of 95%. Slides were examined and photographed with an optical microscope (Nikon eclipse E600) associated with a microphotography system (Nikon DMX1200 digital camera). 

### 2.6. Real-Time Reverse-Transcription Polymerase Chain Reaction (RT-PCR) Analysis

The total RNA, in each hepatic sample, was extracted using the RNeasy Mini Kit (Qiagen, Venlo, The Netherlands) according to the manufacturer’s instructions. Then, RNA was converted to cDNA using QuantiTect Reverse Transcription Kit (Qiagen). The reaction mixtures, containing 10μL of Sybr Green Master Mix 2X (Bio-Rad, Hercules, CA, USA), 1 μM (0.4 μL) of primers for each gene (listed in Table 2) [22], 1 μL of cDNA template (1 μg) in a final volume of 20 μL, were placed in duplicate in wells of a 96-well real-time PCR plate (Bio-Rad). The PCR reaction was performed using a CFX96 Touch Real-Time PCR (Bio-Rad). Thermal cycle profile was as follows: 50 °C for 2 min, 95 °C for 10 min, 95 °C for 15 s (40 cycles), and 60 °C for 1 min, as described by Hacariz et al. [23]. Relative quantitation was achieved by the comparative ∆∆ cycle threshold method, and data were normalized to glyceraldehyde-3-phosphate dehydrogenase [GAPDH] mRNA level and expressed as a fold change compared with controls.

### 2.7. Statistical Analysis

Statistical analysis was performed using GraphPad (version 5.03; GraphPad Software Inc., La Jolla, CA, USA). One-way analysis of variance was used to compare the positive labeling of the different immune cells’ phenotypes among the different groups (A, B, and C). T-tests for two samples assuming unequal variances were used as post hoc tests. Data acquired from Real-Time PCR were analyzed with StatView software (Abacus Concepts, SAS Institute Inc., Cary, NC, USA) by Student’s t-test. Bars represent the mean ± SD (standard deviation) of four independent experiments. For all experiments, *p* < 0.05 was considered statistically significant.

## 3. Results

### 3.1. Gross Examination

Out of 100 total cases, 15 livers showed no relevant pathologic alterations and were negative to all metacestodes and other hepatic parasites (Figure 1a). In contrast, 70 showed only CE lesions and 15 resulted positive exclusively for *Taenia hydatigena* and, thus, were excluded from the study. Macroscopically, livers with fertile hydatid cysts showed irregularly nodular, oval to round, fluid-filled, gray–white areas ranging in size from 1 to 10 cm (Figure 1b). At cut section, fertile cysts were characterized by two whitish walls of approximately 0.5–1 mm: an external one with dense fibrous tissue (fibrous pericyst) and an Internal, more friable one (germinal layer). Such membranes delimited an irregularly shaped cavity, sometimes multilocular with several intercommunicating cystic lesions. Such spaces contained citrine-colored liquid (hydatid liquid) and yellowish-white granular material (hydatid sand). Sterile and regressed cysts evident on the surface of the organ appeared as irregularly nodular, whitish lesions with dimensions ranging from 1 to 15 mm (Figure 1c). At cut section, they were characterized by a whitish wall of 0.5–1 mm consisting of dense fibrous tissue (fibrous membrane) delimiting a small cavity sometimes containing abundant unorganized granular whitish material (calcified material). A predisposition of localization of cysts between the hepatic lobes was not observed. According to the macroscopically examination, livers have been assigned to three different groups: (1) Group A, normal liver (no. 15); (2) Group B, liver with fertile hydatid cysts (no. 28; 40%); and (3) Group C, liver with sterile hydatid cysts (no. 42; 60%).

### 3.2. Histological Examination

Morphological analysis with hematoxylin and eosin staining confirmed the fertility of the cysts evaluated macroscopically. Livers from Group A (control group) did not show relevant pathologic lesions. The hepatic parenchyma of animals from Group B (livers with fertile cysts) was expanded and effaced by multilocular round to irregularly oval hydatid cysts that were surrounded by irregularly thick fibrous bands (Figure 2a). Fibrous connective septa replaced or compressed adjacent hepatic parenchyma that was infiltrated by a mild to moderate, multifocal to coalescing, granulomatous inflammatory population consisting mostly of lymphocytes, plasma cells, macrophages, multinucleated giant cells and rarer eosinophils, and viable and not viable neutrophils (Figure 2b). Parasitic cysts were lined by a thick, eosinophilic outer laminated layer and an inner germinal epithelial layer. Multiple protoscolices were found free within the lumen of the cysts or budding from the germinal layer. Protoscolices showed a thick tegument and a spongy parenchyma containing calcareous corpuscles, suckers, and a rostellum equipped with birefringent hooks. Multifocally, hepatocytes of the adjacent hepatic parenchyma showed shrunken hypereosinophilic cytoplasm and nuclear pyknosis (necrosis). Portal and fibrotic areas adjacent to the hydatid cyst had increased numbers of small bile duct profiles (ductular reaction). Connective tissue surrounding the hydatid cyst was stained blue with Masson’s trichrome staining, showing that it was rich in fibrous elements (Figure 2c). Livers of animals from Group C (sterile cysts) showed a moderate to severe, chronic, granulomatous, multifocal to coalescing inflammatory infiltrate consisting mostly of macrophages, multinucleated giant cells, lymphocytes, plasma cells and rarer eosinophils, and viable and not viable neutrophils. The inflammatory infiltrate was sometimes multifocally centered on necrotic areas admixed with caseous or calcified material (Figure 2d,e). Several scattered hepatocytes showed vacuolar degeneration and pycnotic nuclei. The periportal area showed thickening with increased cellularity. Biliary ductal reaction was also observed. In Masson’s trichrome staining, a blue-stained fibrous connective was observed (Figure 2f). 

### 3.3. Immunohistochemical Evaluation

The inflammatory cell phenotypes were identified based on the staining pattern of antibodies against cell surface proteins. Livers of Group A (control group) showed rare, scattered immunopositivity for T cell (CD3+), B cell (CD20), histiocytes (Iba1+), and absence of immunopositivity for TGFβ and MMP9 antibodies. Immunohistochemical analysis of livers from Group B and Group C revealed immunolabeling of CD20+ B lymphocytes and CD3+ T lymphocytes as densely accumulated clusters around the hydatid cysts (Figure 3). Moreover, diffuse immunolabelling of histiocytes for Iba-1, TGF-β, and MMP9 was observed around the hydatid lesion. No immunoreaction was observed in negative control sections where the primary antibody was omitted (Figure 3). In livers from both Group B and Group C, the number of CD20+ B lymphocytes was higher than CD3+ T lymphocytes (*p* < 0.05) (Figure 4a). The number of inflammatory cells was statistically significantly higher in Groups B and C compared to Group A (*p* < 0.001) (Figure 4a). However, no statistically significant differences were observed between Group B and Group C regarding the number of inflammatory cells (Figure 4a). 

### 3.4. Cytokine Gene Expression Levels

The results of cytokine gene expression levels are shown in Figure 4b. No inflammatory markers are present in Group A. No significant statistical differences in the expression levels of Th-1 immune cytokines INF-γ, TNF-α, and IL-12 were observed among the groups. However, there was a significant increase in expression levels of Th-2 immune cytokines TGF-β and IL-10 in Group B and Group C compared to Group A (*p* < 0.001).

## 4. Discussion

Cystic echinococcosis is endemic in several European countries that surround the Mediterranean basin [1,14,15,25]. The life cycle of the parasite is preserved by the release of *E. granulosus* eggs into the environment by the definitive host (dog and other canids) and the ingestion of these eggs by intermediate hosts where the larval stage of the parasite (hydatid cysts) can survive in tissues and organs for very long periods, often causing chronic infection [10,11]. The hydatid cyst is considered established once the development of the germ epithelium and laminate layers has been completed and once growth to the definitive position in the target tissue has begun [26]. To reach this evolutionary stage, the parasite has to overcome the initial attack of the immune system (in particular, the complement-dependent or antibody-mediated killing) and can continue its development up to the fertile stage. Despite the capacity of the intermediate host to produce a significant immune response against *E. granulosus* infection, the parasite has developed highly effective strategies to elude the host defenses and escape clearance [7,11]. In naturally infected intermediate hosts, there are two kinds of hydatid cysts: the fertile cysts, characterized by protoscoleces free into the hydatid fluid or attached to the germinal layer; and the sterile cysts, characterized by the absence of protoscoleces [27,28]. The reason why these two different kinds of cysts exist is still unclear [28,29], but the host immune response likely participates in generating sterile *Echinococcus* cysts [27]. To our knowledge, studies investigating the local immune response to established tissue cysts in the ovine host are still lacking [13]; thus, the present study aimed to further characterize the inflammatory cytokine production and the cellular inflammatory populations in hepatic, chronic ovine CE and their relationship with cysts fertility. Our findings showed a severe, locally extensive, granulomatous chronic inflammatory infiltrate surrounding both fertile and sterile hydatid cysts. The inflammatory infiltrate comprised mostly mononuclear cells, lymphocytes, and plasma cells. Interestingly, unlike what is generally associated with parasitic infections, eosinophilic granulocytes were rarely observed in the cases examined. Inflammatory infiltrate was diffusely distributed around the parasitic cysts, sometimes in a pseudo-follicular fashion (pseudo-follicular lymphoid structures), testifying to the presence of a long-standing antigenic stimulus [30]. Immunohistochemical results showed that the inflammatory infiltrate predominantly consisted of Iba-1-positive macrophages and that, among lymphocytes, they were generally more represented by B lymphocytes than T lymphocytes. Livers from control groups did not show relevant pathological alteration or the presence of inflammatory infiltrate and inflammatory cytokines expression. Conversely, an important increase in inflammatory cells and cytokine expression was observed in hepatic tissue from animals affected by CE. Surprisingly, no statistically significant differences were observed for the inflammatory infiltrate severity and phenotype and for cytokines’ mRNA levels between Group B (fertile cysts) and Group C (sterile cysts). In this study, we can surely confirm that macrophages are mostly involved in the local immune response to hydatid cysts in sheep [5]. Macrophages are a vital component and powerful effector cells of the innate immune system, playing an essential role in inflammation, host defense and embryonic development, removal of cellular debris, and tissue repair. Conventionally, two main macrophage activation pathways are recognized: classical (M1) and alternative (M2). Macrophages activation (M1 vs. M2) is a pathway that defines the different functional phenotypes adopted by macrophages in response to specific signals from the microenvironment. Specifically, M1 macrophages have been recognized as a pro-inflammatory type engaging in the direct defense against pathogens and the production of pro-inflammatory cytokines and microbicidal molecules. Conversely, M2 macrophages have been recognized as having the opposite function, consisting of inflammation resolution and tissue repair [5]. For this study, we did not perform a specific characterization of M1 and M2 phenotypes; hence, we can only speculate that the different type of macrophages activation may be related to the presence of fertile or sterile cysts. However, a very important and well-designed study by Atmaca suggested that both M1 and M2 phenotypes are involved in the local immune response to sterile and fertile hydatid cysts and that Th1 and Th2 immune reaction stimulation persists together [5]. Our results suggest that Th2 immune response may be dominant in CE, confirming that B cells are extremely important for the control of the immune response during parasitic infection [31,32] even though the regulation of B cells response in *E. granulosus* infection is still a subject for discussion. Several authors have recently established that B cells may negatively regulate immune responses, hence the novel definition of *regulatory B cells* (Breg or B10 cells) [32,33]. Regulatory B cells may evoke several IL-10-dependent regulatory effects that include downregulation of proinflammatory cytokines, induction of Treg cells, and production of TGF-β [32,34,35,36,37]. In this study, the presence of TGF-β-positive macrophages and the increase in IL-10 and TGF-β mRNA levels were the prominent characteristics of the inflammatory line of defense developed against the parasite in both livers with fertile and sterile hydatid cysts. IL-10 and TGF-β are abundantly expressed in leukocytes in CE-infected hosts, especially in the immediate vicinity of the parasite [26,36], possibly playing an important immunomodulatory role in ensuring the persistence of the parasite within the host [38,39,40]. This hypothesis is supported by experimental studies suggesting that the establishment of a polarized type-2 cytokine in response to non-proteic antigens and the early secretion of IL-10 by B cells may favor local immunosuppression and permit parasite survival [41,42,43]. Similarly, it has been established that IL-10 and TGF-β modulate the immunologic mechanisms by which macrophages both destroy the parasite and repair tissue damage caused by the parasite [5,44]. Regarding the results of IFN-γ and IL-12, we did not observe a relevant expression of these cytokines nor statistically significant differences among the groups. IFN-γ is a key cytokine for the inhibition of the growth and function of helminths and other infectious agents through the stimulation of nitric oxide (NO) production by macrophages [45]. It has been shown that IFN-γ is also capable to enhance the production of IL-12, establishing a protective Th1-mediated immunity during *E. granulosus* infection [45,46,47,48]. IL-12 is produced mainly by activated macrophages/monocytes and has a very important role in the initiation and regulation of innate cellular immune responses [28,49]. IL-12 serum level is generally increased in patients with hydatidosis [40], and the effect of IFN-γ and IL-12 on the hydatid cyst has been also verified in vitro [45]. Moreover, a study from Amri et al., revealed that hydatid cyst fluid contains IL-12 and that the quantity of this cytokine is noticeably higher in the fertile cysts compared to the sterile ones [40]. With their work, Amri et al., suggested that protoscoleces’ excretory/secretory components allegedly play a crucial role in the stimulation of the immune system by leading to IL-12 production. Our results are not in agreement with these observations from human or bovine infections as CE-infected animals did not show significant levels of IFN-γ and IL-12 [28,50]. Nonetheless, our findings are similar to the study described by Fardinia et al. [51], which did not observe differences in IL-12 expression between normal and infected sheep. These data need to be further investigated, but we can speculate that there are several differences in the cellular immune response between ovine and other species, such as humans or cattle. Moreover, a decrease in IFN-γ and IL-12 during chronic CE in sheep may be because of the shift from Th1 to the Th2 subtype [9]. Finally, we observed a higher expression of MMP9-immunolabelled macrophages in both livers with fertile and sterile hydatid cysts compared to healthy hepatic tissue. To our knowledge, this is the first study to investigate the role of MMP9 in the development of liver fibrosis in sheep naturally infected by CE. Our results suggest that the progression of liver fibrosis in *E. granulosus* infection could be associated with upregulated expression of MMP9 [52]. Since liver fibrosis may cause a severe chronic disease, estimating the presence and severity of liver damage may be critical for treatment. Our results may potentially be of interest to future research aimed to validate the evaluation of serum markers for staging liver fibrosis. Collectively, our data suggest that, in ovine hepatic CE, a strong Th2 response was associated with both fertile and sterile cysts. Further investigations are required to confirm these results and to better explore the local inflammatory response involved in the different stages of CE in sheep.

## 5. Conclusions

To date, there is little data on the local inflammatory response associated with *E. granulosus* infection in the ovine liver. We hope that the present study can significantly contribute to further identifying the immunological mechanisms involved during the different stages of the development of ovine hydatidosis. In our opinion, extensive knowledge of the immunopathology of echinococcosisis is necessary for the implementation of strategies aiming at the diagnosis, prevention, and therapy of this disease in sheep as well as other animals and humans. For example, a valuable approach could be the administration of cost-effective vaccines or potential target antigens that can induce, maintain, or re-orientate the host immune response at a Th1-level, inhibiting or restricting metacestode growth [52]. This kind of “active immunotherapy” has to be further refined in sheep because apparently there are some differences between ovine and other species, including humans. However, the continuous developments of the “omics” sciences (genomics, proteomics, metabolomics) may reveal important similarities between parasite–host interplay in echinococcal infections in different animal species and humans, also providing new spontaneous animal models for the identification of additional targets for diagnosis, vaccination, and therapy.

## Figures and Tables

**Figure 1 vetsci-10-00315-f001:**
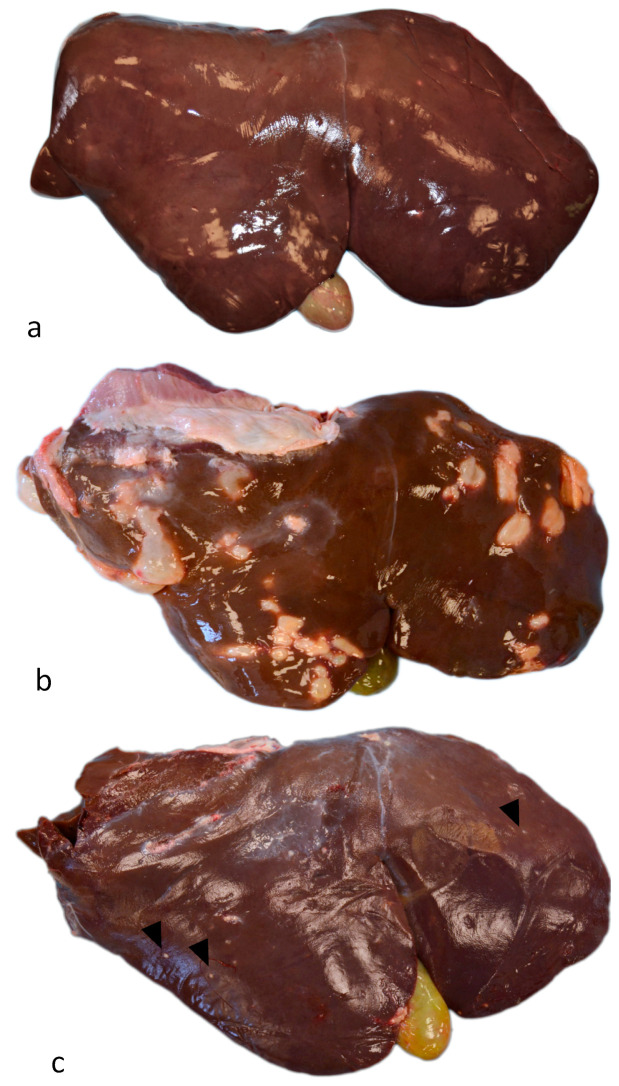
Sheep, liver, cystic echinococcosis. (**a**) Normal liver. (**b**) Liver with fertile hydatid cysts: numerous coalescing, whitish, and protruding multifocal lesions are evident on the surface of the organ. (**c**) Liver with sterile, regressed, and calcified cysts: several disseminated, small, and slightly protruding whitish lesions are evident on the surface of the organ (arrowheads).

**Figure 2 vetsci-10-00315-f002:**
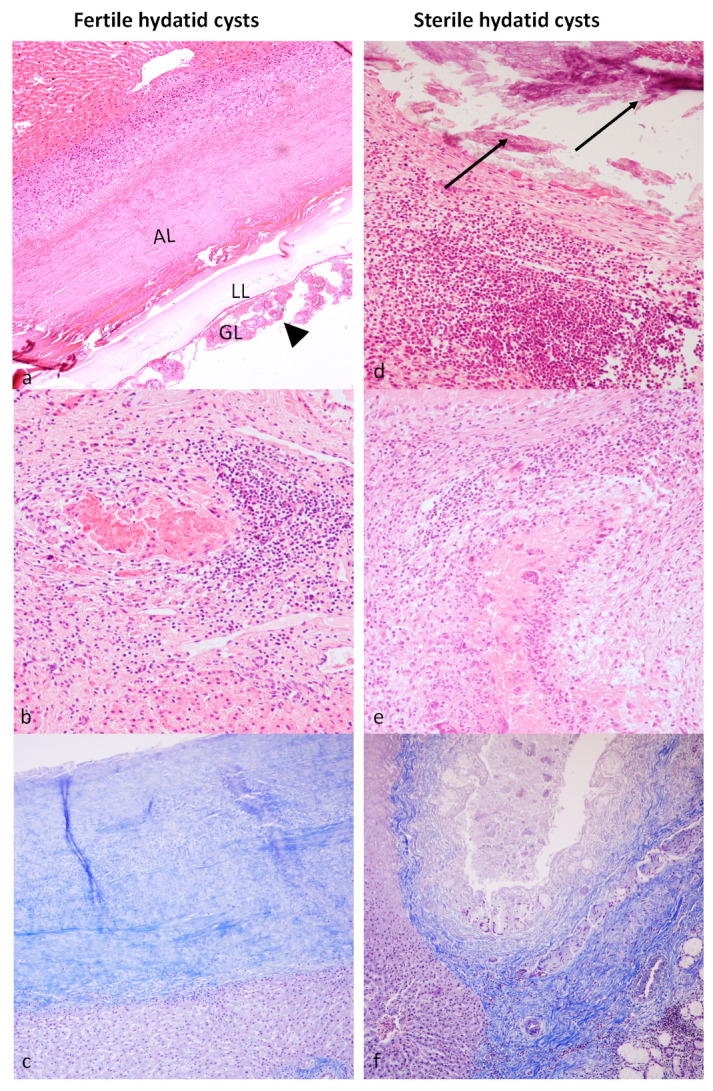
Hydatid cysts, liver, sheep. (**a**) Fertile cyst. It is possible to distinguish the adventitial layer (AL) surrounded by a severe and diffuse inflammatory infiltrate, the laminated layer (LL), and the germinal layer (GL) with numerous protoscolices (arrowhead). (**b**) Severe, chronic, granulomatous inflammatory infiltrate and hepatocytes necrosis. (**c**) The adventitial layer is stained blue with Masson’s trichrome. (**d**) Sterile and regressed cyst. The fibrous layer is surrounded by a severe and diffuse inflammatory infiltrate and surrounds abundant mineralized material (arrows). (**e**) Inflammatory infiltrate is centered on necrotic areas admixed with caseous or calcified material. In Masson’s trichrome staining (**f**), blue-stained fibrous connective tissue is clearly observed. Hematoxylin–eosin for a, b, d, and e. Masson’s trichrome staining for c and f. Original magnification 20×.

**Figure 3 vetsci-10-00315-f003:**
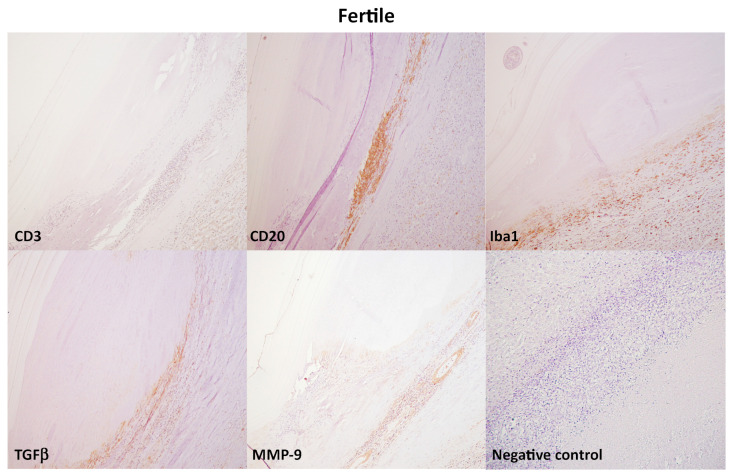
Immunohistochemical characterization of inflammatory infiltrate, liver, sheep. Immunohistochemical analysis of livers from Group B and Group C revealed immunolabeling of CD3+ T lymphocytes and CD20+ B lymphocytes as densely accumulated clusters around the hydatid cysts. Moreover, diffuse immunolabelling of histiocytes for Iba-1, TGF-β, and MMP9 was observed around the hydatid lesion. No immunoreaction was observed in negative control sections where the primary antibody was omitted. 3,3 -diaminobenzidine (DAB) chromogen, hematoxylin counterstain. Original magnification, 20×.

**Figure 4 vetsci-10-00315-f004:**
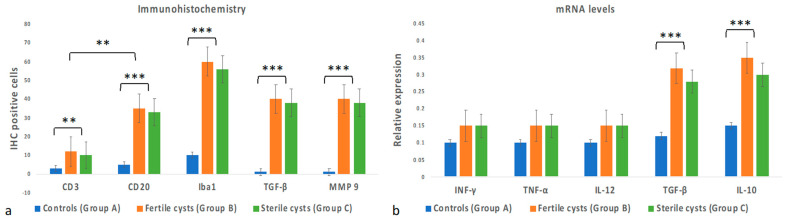
Changes in inflammatory infiltrate and cytokines expression between the groups. (**a**) In livers from both Group B and Group C, the number of CD20+ B lymphocytes was higher than CD3+ T lymphocytes. The number of inflammatory cells was statistically significantly higher in Groups B and C than in Group A. No statistically significant differences were observed between Groups B and C regarding the number of inflammatory cells. (**b**) No inflammatory markers are present in Group A. The expression levels of Th-1 immune cytokines INF-γ, TNF-α, and IL-12 showed no statistically significant differences among the groups. However, there was a significant increase in expression levels of Th-2 immune cytokines TGF-β and IL-10 in Groups B and C compared to Group A. Each value is the mean ± SEM (** *p* < 0.05 vs. control; *** *p* < 0.001). SEM, standard error of the mean.

**Table 1 vetsci-10-00315-t001:** Immunohistochemical protocols and primary antibodies used for cellular type characterization.

Antibody	Specificity	Epitope Demasking	Dilution	Reference
CD3 (IS503, rabbit polyclonal antibody, DAKO).	Pan T cell marker	Citrate pH 6, 20 min	1:200	[18,20]
CD20 (ACR3004B, rabbit polyclonal antibody, Biocompare)	Pan B cell marker	Citrate pH 6, 20 min	1:50	[20]
Iba-1 (019_19741, rabbit polyclonal antibody, WAKO).	Macrophages	Citrate pH 6, 20 min	1:800	[20]
TGFβ (ab9758, rabbit polyclonal antibody, AbCam)	Macrophages	Citrate pH 6, 20 min	1:200	[21]
MMP-9 (ab38898, rabbit polyclonal antibody, Abcam).	Macrophages, Fibroblasts	Citrate pH 6, 20 min	1:200	[21]

**Table 2 vetsci-10-00315-t002:** List and sequences of the primers for ovine cytokine and housekeeping genes used.

Ovine Gene Target	Primer Sequences	Reference
TNF-α	Forward: GGTGCCTCAGCCTCTTCTC	[23]
	Reverse: GAACCAGAGGCCTGTTGAAG	[23]
INF-γ	Forward: CAAATTCCGGTGGATGATCTG	[24]
	Reverse: GCGACAGGTCATTCATCACCTT	[24]
IL-12	Forward: TCTCGGCAGGTGGAAGTCA	[23]
	Reverse: ACTTTGGCTGAGGTTTGGTCTG	[23]
IL-10	Forward: CCAGGATGGTGACTCGACTAGAC	[24]
	Reverse: TGGCTCTGCTCTCCCAGAAC	[24]
TGF-β	Forward: AAGCGGAAGGGCATCGA	[24]
	Reverse: CGAGCCGAAGTTTGGACAAA	[24]
GAPDH	Forward: GGCGTGAACCACGAGAAGTATAA	[23]
	Reverse: CCCTCCACGATGCCAAAGT	[23]

## Data Availability

All data used in the current study are available from the corresponding author on reasonable request.

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
