# Peer review of "Evaluation of the Local Immune Response to Hydatid Cysts in Sheep Liver"

_vetsci, 2023, doi:10.3390/vetsci10050315_

Round 1

Reviewer 1 Report

The manuscript evaluates the local immune response to natural cystic echinococcosis (CE) in sheep livers using immunohistochemistry and RT-PCR. The manuscript is well written and the number of cases included is adequate. The manuscript is original and the results are novel and interesting in the field of host-parasite interaction. However, before publication some aspects of the manuscript should be improved:

1.- In section Material and methods, some more details about the primary antibodies should be given, ie. Was applied some antigen retrieval method for some of the antibodies used?. The specificity of the antibodies should be given and information of cross reactivity with ovine antigens should be provided. Basic information of functions of the antigen detected by some antibodies would also be useful to readers, particularly those for Iba1 and MMP9.

2.- In section Results the lesions in figure 3c should be pointed since they are small and difficult to appreciate.

3.- In section Discussion some information about the Iba1 antibody should be included. It is surprising that none of the antibodies or cytokines detected by RT-PCR showed statistical differences between the fertile and sterile hydatid cyst groups. This merit some discussion, ie. If the type of macrophage activation (classical or alternative) may be related to fertile or sterile cysts.

Author Response

We would like to express our great appreciation for the nice comments and the extremely valuable suggestion on our manuscript entitled “Evaluation of the local immune response to hydatid cysts in sheep liver”. All of your comments were very helpful for revising and hopefully improving our paper. We have studied these comments carefully and have made corresponding corrections that we hope will meet with your approval. The responses to your comments are provided below; specifically, the questions are highlighted in bold and our responses are in italics. The changes in the revised manuscript are tracked. 

The manuscript evaluates the local immune response to natural cystic echinococcosis (CE) in sheep livers using immunohistochemistry and RT-PCR. The manuscript is well written, and the number of cases included is adequate. The manuscript is original, and the results are novel and interesting in the field of host-parasite interaction. However, before publication some aspects of the manuscript should be improved:

1.- In section Material and methods, some more details about the primary antibodies should be given, ie. Was applied some antigen retrieval method for some of the antibodies used? The specificity of the antibodies should be given and information of cross reactivity with ovine antigens should be provided. Basic information of functions of the antigen detected by some antibodies would also be useful to readers, particularly those for Iba1 and MMP9. 

We revised Table 1 adding antigen retrieval methods, cell target of the primary antibody and also a reference of previous peer-reviewed and published study that validated the antibodies used for our experiments.

2.- In section Results the lesions in figure 3c should be pointed since they are small and difficult to appreciate. 

We agree with the reviewers; hence, we modified the figures accordingly with your valuable suggestion.

3.- In section Discussion some information about the Iba1 antibody should be included. It is surprising that none of the antibodies or cytokines detected by RT-PCR showed statistical differences between the fertile and sterile hydatid cyst groups. This merit some discussion, ie. If the type of macrophage activation (classical or alternative) may be related to fertile or sterile cysts.

Thank you for your comment that allowed us to further discuss this result. We added a paragraph in the Discussion according to reviewer concern.

Thanks again for your comments...we hope that our manuscript is now much more improved and suitable for publication on “Veterinary Science“Journal.

Paola Pepe

Reviewer 2 Report

The submitted manuscript entitled „Evaluation of the local immune response to hydatid cysts in sheep liver” described a biochemical aspects of host-pathogen interaction after infection of the intermediate ovine host with the parasite - Echinococcus granulosus. This is a valuable manuscript and it may contribute to the implementation of strategies for the reduction of the prevalence of infection in sheep and identify the immunological mechanisms involved during the different stages of development of ovine hydatidosis. E. granulosus is a potential zoonotic pathogen, so this article may offer ideas for immunotherapy strategies in human cystic echinococcosis as well.

Article is well constructed, materials and methods are well described and developed. Results are clearly presented. However, there are some stylistic and punctuation errors and I suggest that it need some minor corrections and text editing before publication. Please find comments and suggestions in the pdf file.

Author Response

We would like to express our great appreciation for your nice comments and the extremely valuable suggestion on our manuscript entitled “Evaluation of the local immune response to hydatid cysts in sheep liver”. All of your comments were very helpful for revising and hopefully improving our paper. We have studied these comments carefully and have made corresponding corrections that we hope will meet with your approval. The changes in the revised manuscript are tracked. 

We hope that our manuscript is now much more improved and suitable for publication on “Veterinary Science“Journal.

Kind regards,

Paola Pepe

Reviewer 3 Report

The study was comprehensively designed and clearly explained.  The findings aimed to characterize the inflammatory phenotype in ovine liver cystic echinococcosis by the evaluation of different markers related to the local immune response. The results show that macrophages have a predominant role in the local immune response to hydatid cysts. Moreover, that’s allow to authors speculate that Th2 immune response may be dominant, confirming that B cells are decisively essential in the control of the immune response during parasitic infection and that the immunomodulatory role of IL-10 and TGF-b may ensure the persistence of the parasite within the host. The results of the present study may contribute to identify the immunological mechanisms involved during the different stages of development of ovine hydatidosis.

Author Response

Thank you for your very nice comments.

Reviewer 4 Report

The paper describe a study aimed to describe the inflammatory response against Echinococcus granulosus cysts in ovine liver

The paper is interesting and well written

Some remarks should be keep in account.

Simple summary:

Line 19: “of human and herbivores (intermediate hosts)”: this sentence should be slightly remodeled as “of herbivores/omnivores (intermediate hosts) including human”. This because animal herbivores (but also omnivores) can act as intermediate host, human act as accidental intermediate host that do not contribute to the parasitic cycle.

Simple summary and abstract should be revised because, in this case, simple summary actually act as the introduction of the abstract.

Results

Line 200: Livers negative for metacestodes (control group) it is stated that showed “no relevant pathologic alteration”. This in my opinion is correct, because in sheep very frequently the livers, even without presenting metacestodes, show alterations, and are often affected by Dicrocoelium. Was the presence of Dicrocoelium observed in this case, even in the livers with metacestodes? Did the collection of liver samples take this into account, avoiding particularly parasitized areas?

Figure 1: the single figures are very little, and it's a bit difficult to observe them well, but it seems that figure 1A is inverted with figure 1C

Line 238: only a sucker? Or the suckers?

Figure 2: the figures are very small. I would recommend placing the series refer to sterile and fertile hydatid cysts vertically, so that for each row only two photos are placed side by side and can be enlarged.

Figure 3: as in figure 2. (in particular in figure fertile/CD3 nothing is distinguished)

Conclusion

The conclusion should be improved.

In my opinion this study contributes to the identify the immunological mechanisms involved during the different stages of development of ovine hydatidosis, but:

1)            specify better as it may contribute to the implementation of strategies for the reduction of the prevalence of infection in sheep (perhaps vaccines capable of directing towards a Th1 response?);

2)            as stated in line 381-383, seems to be some differences in cellular immune response between ovine and other species, such human or cattle, so better clarify how this study on sheep can give ideas for immunotherapy strategies in human cystic echinococcosis.

Author Response

We would like to express our great appreciation to you for the nice comments and the extremely valuable suggestion on our manuscript entitled “Evaluation of the local immune response to hydatid cysts in sheep liver”. All of your comments were very helpful for revising and hopefully improving our paper. We have studied these comments carefully and have made corresponding corrections that we hope will meet with your approval. The responses to the reviewers’ comments are provided below; specifically, the questions are highlighted in bold and our responses are in italics. The changes in the revised manuscript are tracked. 

The paper describes a study aimed to describe the inflammatory response against Echinococcus granulosus cysts in ovine liver 

The paper is interesting and well written.

Some remarks should be kept in account.

Simple summary:

Line 19: “of human and herbivores (intermediate hosts)”: this sentence should be slightly remodeled as “of herbivores/omnivores (intermediate hosts) including human”. This because animal herbivores (but also omnivores) can act as intermediate host, human act as accidental intermediate host that do not contribute to the parasitic cycle.

Simple summary and abstract should be revised because, in this case, simple summary actually act as the introduction of the abstract. 

Thank you for your comments that were very useful to hopefully improve our manuscript. We modified the simple summary and the abstract according to reviewer comments.

Results

Line 200: Livers negative for metacestodes (control group) it is stated that showed “no relevant pathologic alteration”. This in my opinion is correct, because in sheep very frequently the livers, even without presenting metacestodes, show alterations, and are often affected by Dicrocoelium. Was the presence of Dicrocoelium observed in this case, even in the livers with metacestodes? Did the collection of liver samples take this into account, avoiding particularly parasitized areas?

Thank you for your comment that allowed us to add an important information within the manuscript. The presence of liver parasites (i.e. Dicrocelium or Fasciola) and related lesions was of course investigated for each liver. Liver from control group didn’t show any significant pathologic alterations or the presence of hepatic parasites.

Figure 1: the single figures are very little, and it's a bit difficult to observe them well, but it seems that figure 1A is inverted with figure 1C

Line 238: only a sucker? Or the suckers?

We corrected the typo within the text.

Figure 2: the figures are very small. I would recommend placing the series refer to sterile and fertile hydatid cysts vertically, so that for each row only two photos are placed side by side and can be enlarged.

Figure 3: as in figure 2. (in particular in figure fertile/CD3 nothing is distinguished)

We agree with the reviewers; hence, we modified the figures accordingly with your valuable suggestion.

Conclusion

The conclusion should be improved.

In my opinion this study contributes to the identify the immunological mechanisms involved during the different stages of development of ovine hydatidosis, but:

1)            specify better as it may contribute to the implementation of strategies for the reduction of the prevalence of infection in sheep (perhaps vaccines capable of directing towards a Th1 response?);

2)            as stated in line 381-383, seems to be some differences in cellular immune response between ovine and other species, such human or cattle, so better clarify how this study on sheep can give ideas for immunotherapy strategies in human cystic echinococcosis.

We modified the conclusions according to reviewer comments.

Thanks again for your comments...we hope that our manuscript is now much more improved and suitable for publication on “Veterinary Science“Journal.

kind regards,

Paola Pepe